# Estimation of Storm-Centred Areal Reduction Factors from Radar Rainfall for Design in Urban Hydrology

**Søren Thorndahl *** **, Jesper Ellerbæk Nielsen and Michael R. Rasmussen**

Department of Civil Engineering, Aalborg University, DK9220 Aalborg, Denmark; jen@civil.aau.dk (J.E.N.); mr@civil.aau.dk (M.R.R.)

**\*** Correspondence: st@civil.aau.dk; Tel.: +45-9940-8475

**Abstract:** In the design practice of urban hydrological systems, e.g., storm-water drainage systems, design rainfall is typically assumed spatially homogeneous over a given catchment. For catchments larger than approximately 10 km$^2$, this leads to significant overestimation of the design rainfall intensities, and thus potentially oversizing of urban drainage systems. By extending methods from rural hydrology to urban hydrology, this paper proposes the introduction of areal reduction factors in urban drainage design focusing on temporal and spatial scales relevant for urban hydrological applications (1 min to 1 day and 0.1 to 100 km$^2$). Storm-centred areal reduction factors are developed based on a 15-year radar rainfall dataset from Denmark. From the individual storms, a generic relationship of the areal reduction factor as a function of rainfall duration and area is derived. This relationship can be directly implemented in design with intensity–duration–frequency curves or design storms.

**Keywords:** areal reduction factor; areal rainfall; design storms; urban drainage

## 1. Introduction

Design of simple urban drainage systems is traditionally based on rainfall intensity–duration–frequency (IDF) relationships [1,2], which also is the case in Denmark [3], where it is not common practice to comprise areal rainfall in the design of drainage systems. Consequently, as urban catchments continue to increase in size and wastewater treatment plants (treating both waste- and stormwater) are centralised, areal rainfall cannot always be neglected in design practice without compromising design recommendations. If the spatial component is neglected, systems might end up being oversized leading to excessive costs.

Traditionally, areal reduction factors (*ARF*) are estimated correlating multiple rain gauge recordings in pairs and deriving spatial correlations for different distances between rain stations/gauges [4–6]. They can also be estimated empirically as the ratio between maximum areal rainfall (spatially averaged rain gauge data) and point rainfall over specific durations within a fixed area [7–13]. The past decade's development in weather radar rainfall shows the potential to derive the relationships from this type of data instead. Compared to rain gauge networks, radar data adds a significant advantage for the areal coverage. Uncertainties related to spatial interpolation between point stations from rain gauge data are, thus, avoided. Development of areal reduction factors from radar data has been presented in several studies [14–19].

*ARFs* can be divided into two general classes as presented in the extensive review by Svensson and Jonas [20]: (i) The storm-centred approach in which the *ARF* is derived searching for the maximum rainfall intensity in a given domain and estimating the ratio between areal and point rainfall individually storm by storm [21]; and (ii) the geographically-fixed or area-fixed approach, in which the ratio between

maximum areal rainfall with a given return period and maximum point rainfall with the same return period is derived at a fixed location and linked to the extreme event rainfall statistics of this point [21]. As argued by Wright et al. [18], the maximum areal rainfall and the maximum point rainfall might not originate from the same storm, causing the area-fixed approach to be a ratio of statistical values rather than values representing the actual spatial variability of rainfall. The area-fixed approach can be considered a statistical approach, whereas storm-centred is an empirical approach. The storm-centred approach has been criticised [5,21] because it does not depend on the return period of design rainfall. On the other hand, Wright et al. [18] argue that the area-fixed *ARFs* are not valid for the "true" properties of rainfall, since the statistical approach mixes observations from different storms and storm types, creating a discrepancy between recurrences of maximum point rainfall and areal rainfall. Wright et al. [18] also argue that it is not possible to identify any systematic difference in statistics for smaller or larger storms and thus, concluded an independence of *ARF* and return period for their case study area. Furthermore, they recommend studying variability in the relationship between point rainfall and areal rainfall in more detail before incorporation into designs of hydrological systems.

Proceeding in the line with the conclusions of Wright et al. [18], the main objective of this study is to estimate areal reduction factors based on the storm-centred approach and to derive a methodology for considering areal rainfall in design of urban drainage systems—thus focusing on short rainfall durations (one minute to one day) and areas smaller than 100 km$^2$ [22–24]. In comparison with other studies, the focus at the small scales in space and time is novel.

This study uses a radar rainfall dataset from Denmark covering 15 years to validate the proposed methodology. As detailed above, a few other authors have developed areal reduction factors based on radar rainfall data. Some of these studies are applied in a context of rural or natural waterways, and some address the spatial rainfall variability for urban catchments. Still, there is a need for further research in the small scale variability of rainfall in time and space in a context of urban design applications [11,23–25].

The application of the derived methodology is limited to the design of urban drainage systems since they are of the main motivation of the study, and a main research area of the authors [26–28]. The methodology, however, can serve in a broader application within hydrological systems, where small time and space scales are of interest.

The paper is structured as follows: The radar rainfall data set is presented in Section 2.1 along with a methodology for correction of pixel scale error in Section 2.2. In Section 2.3, a novel method for calibrating a generic 3-parameter relationship of the *ARF* as a function of area and rainfall duration (aggregation level) is developed. Results are presented in Section 3 and discussed in Section 4 along with a comparison of derived *ARF* values with literature equivalents in Section 4.1; reflections on the implementation of the derived *ARF* in urban hydrological design are presented and discussed in Section 4.2. Finally, conclusions are given in Section 5.

## 2. Materials and Methods

### 2.1. Data

Details of radar rainfall data presented in this paper are described in Thorndahl et al. [29] covering a period from 2002 to 2012. The dataset is expanded with an additional 4 years to cover the period from 2002 to 2016. The raw radar data is measured with a single C-band Doppler radar (manufactured by EEC, Enterprise, AL, USA) and operated by the Danish Meteorological Institute (DMI). The total range of the radar is 240 km, and the 100 km range will be used for quantitative precipitation estimates. This range covers an area of ~31,400 km$^2$, including the greater Copenhagen area and the island of Sealand in Denmark as well as parts of south-western Sweden (Figure 1). The spatial resolution of the cartesian rainfall product is $500 \times 500$ m$^2$ generated in a 1 km altitude pseudo-CAPPI layer. This product is selected due to consistency in data having a fixed elevation as a function of range and significantly fewer errors and clutter compared to lower altitude pseudo CAPPI products. The temporal resolution

of the original dataset is 10 min, but the developed dataset has been regenerated by using a mixed forward and backward advection interpolation method by Nielsen et al. [30] to a 1 min resolution. Radar reflectivity is converted into rainfall intensities using a fixed Marshall–Palmer relationship [31] and bias-adjusted against with 67 rain stations/gauges using a daily mean field bias adjustment approach as described in Thorndahl et al. [30] and Smith and Krajewski [32]. As documented in Thorndahl et al. [29], the mean-field bias adjustment is applied within the 100 km range domain of the radar on a daily time scale. Regional variability of the bias is insignificant and can be neglected [29].

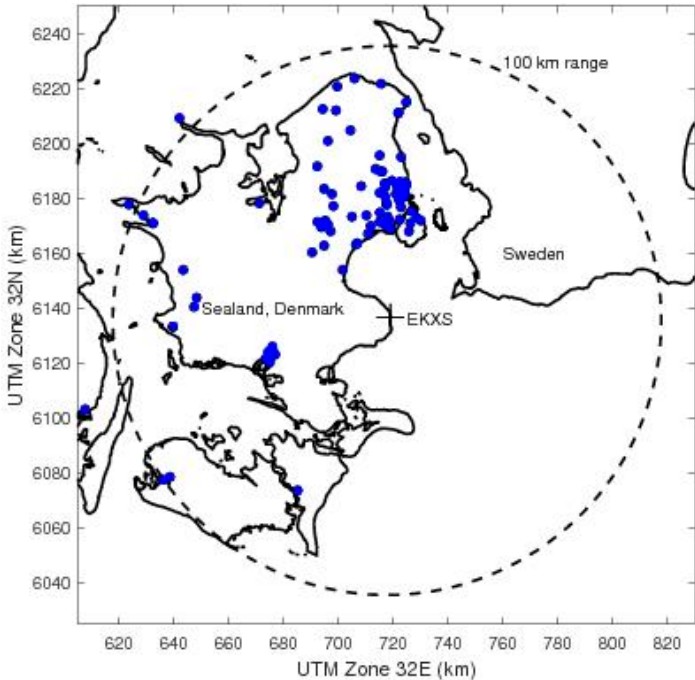

**Figure 1.** Domain of radar (EKXS) rainfall dataset and rainfall stations/gauges (blue dots) used of bias adjustment of radar data.

The rain gauge/radar pairs applied to calculate the daily bias consists of only one rain station/gauge per $500 \times 500$ m$^2$ pixel. For this study, 534 individual rainy days with more than 1 mm of rainfall (in at least one of the rain stations/gauges) were selected. All days with significant noise (defect filters), missing images due to hardware or communication failures, poor bias adjustment (defined as a Nash–Sutcliffe Efficiency below 0, e.g., due to little gauge data, gauge data failures) were omitted. Consequently, the dataset consists of 534 discrete days and is thus discontinuous for the period (2002–2016). In traditional rainfall design statistics of rainfall, an incomplete dataset with periods of missing data is subject to uncertainty in the estimation of return periods. However, since this study disregards estimating return periods, the discontinuous data are not considered any further.

### 2.2. Correction for Pixel Scale Error

In the rain gauge-based development of *ARFs*, the areal component is usually estimated by interpolating over a number of rain stations/gauges using a spatial interpolation technique, such as kriging, inverse distance weighting, or Thiessen polygons. With the use of radar data, it is not necessary to interpolate in space. However, there is a need to know the "true" point rainfall. Even with a fairly high spatial resolution of $500 \times 500$ m$^2$ as applied here, there is a need to acknowledge the scale error between rain gauge and radar pixel [25,33–37]. This representativeness error between rain gauge and radar can originate from several sources. It can be due to rainfall variability itself within a radar pixel meaning that the average rainfall over a $500 \times 500$ m$^2$ is not the same as recorded in a rain gauge with a surface area of less than 0.05 m$^2$. It can also be due to artefacts measuring with radars, such as the

difference between the atmosphere and ground, wind drift, timing errors. Rather than attempting to estimate the rainfall variability at subpixel scale, a data-driven approach for estimation of the mean error is suggested in this paper. This involves calculating a ratio between maximum intensities at different durations (aggregation levels) from rain gauge and radar, respectively. The estimation of the pixel scale error is thus represented through duration-dependent bias factor

$$B(d) = \frac{\sum_{n=1}^{N}\left(\sum_{t=1}^{T}(max(i_G(n,d)))\right)}{\sum_{n=1}^{N}\left(\sum_{t=1}^{T}(max(i_R(n,d)))\right)},$$ (1)

where $i_G$ is the rainfall intensity from rain stations/gauges ($G$) averaged over the duration $d$, and correspondingly $i_R$ is the rainfall intensity averaged over duration $d$ which is calculated from daily mean-field bias adjusted radar ($R$) estimates based on Thorndahl et al. [29]. $n$ is the total number of rain station/gauge radar pixel pairs within the radar range for a specific day $t$. $T$ is the total number of selected rainy days. The *max* function implies that the maximum the daily maximum intensity over duration $d$ is applied in rain gauge data and radar, respectively. The developed biases are presented in the result section.

### 2.3. Method Development

Radar rainfall data from the database are averaged in time and space applying N-dimensional convolution in MATLAB using discrete values of rainfall duration $d$ and area $A$. A principle sketch of the averaging procedure is shown in Figure 2.

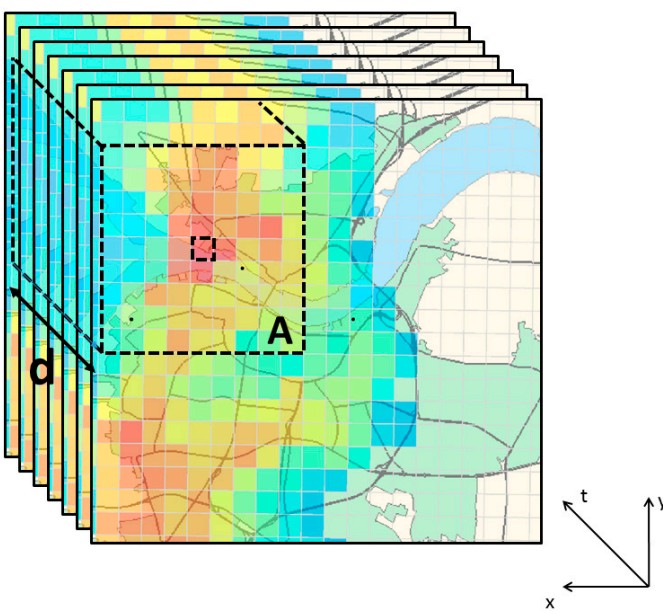

**Figure 2.** Principle of averaging radar images spatially over an area ($A$) and temporally over a duration ($d$).

The areal reduction factor (*ARF*) is defined by the ratio between the maximum radar rainfall intensity ($i_{R,max}$), averaged over duration $d$, area $A$ and the maximum rainfall intensity in a point ($A{\to}0$). Since a storm-centred approach is applied, the maximum point rainfall, $i_{R,max}(d, A \to 0)$ is estimated in the point of the maximum rain intensity within the extent of the averaged area, $A$, (Figure 2). For each rainy day and selected discrete values of $A$ and $d$, the maximum radar rainfall intensity within the domain $i_{R,max}$ is calculated. In this paper, the combination of rainy day, $A$ and $d$ defines a storm, $s$. The estimation of the *ARF* is conducted for a number of individual storms ($s$).

$$ARF(d, A, s) = \frac{i_{R,max}(d, A, s)}{i_{R,max}(d, A \to 0, s)}.$$ (2)

The storm definition allows for multiple locations (within the domain) of the paired areal and point maximum rainfall intensity, depending on duration and area within the same rainy day. It is a key feature of the storm-centred approach that the location is not fixed. A limit of the storm definition is that only one peak (maximum intensity and location of maximum intensity) per storm can be selected.

The ratio of the maximum areal intensity for a given duration with the maximum point intensity over the same duration is calculated. As presented in the data section, the "true" point rainfall is unknown. Instead of the point rainfall, the average rainfall over a pixel size, $p$ is applied. The derived bias Equation (1) between rain station/gauge and radar a function of duration is applied to account for the pixel scale error. The *ARF* can thus be estimated by

$$ARF(d, A, s) = \frac{i_{R,max}(d, A, s)}{i_{R,max}(d, p, s) \cdot B(d)}. \tag{3}$$

To approximate the individual *ARF* relationships to a more generic relationship of the *ARF* as a function of duration and area, recognised methods from the literature are employed in the next sections. Following Rodriguez-Iturbe and Mejía [4] and Sivapalan and Blöschl [5], the *ARF* is assumed to be exponentially dependent on the ratio between the distance between two points and the correlation length, $\lambda$.

$$ARF = \exp\left(-\frac{1}{2} \cdot \left(\frac{A}{\lambda^2}\right)^{\frac{1}{2}}\right), \tag{4}$$

where $\lambda$ is the correlation length and $\frac{A}{\lambda^2}$ is defined as the scaled area.

Other authors (e.g., Villarini et al. [38]) have expanded this relationship to allow for the *ARF* to depend on arbitrary parameters:

$$ARF = \exp\left[-c_1\left(\frac{A}{\lambda^2}\right)^{c_2}\right], \tag{5}$$

where $c_1$ and $c_2$ are coefficients calibrated from record data. Based on the *ARF* as a function of area, $\lambda$ is calibrated for each duration and storm using a non-linear least squares approximation. To limit the degrees of freedom, Equation (4) (i.e., Equation (5) with parameters $c_1 = 0.5$ and $c_2 = 0.5$) is initially applied to fit the *ARF*.

The novelty of this study is to develop a relationship between $\lambda$ and $d$. Preliminary analysis of record values of $\lambda$ for each duration suggests using a power-function approximation described by the following equation:

$$\lambda = a_1 d^{a_2}. \tag{6}$$

Implementing this expression in Equation (5) leads to the following relationship:

$$ARF(A, d) = \exp\left[-c_1\left(\frac{A}{(a_1 d^{a_2})^2}\right)^{c_2}\right]. \tag{7}$$

Applying the relationship developed in Equation (6), the parameters $c_1$ and $c_2$ (Equation (5)) are fitted by the non-linear least squares approximation method. This leads to a modification of the initial values of parameters from Equation (4). In this way, all four parameters are calibrated, and Equation (7) can be simplified to the following relationship:

$$ARF(A, d) = \exp\left[-b_1 \frac{A^{b_2}}{d^{b_3}}\right], \tag{8}$$

where $b_1 = \frac{c_1}{a_1^{2 \cdot c_2}}$, $b_2 = c_2$, and $b_3 = a_2 \cdot c_2 \cdot 2$.

Thus, Equation (8) is developed as a 3-parameter relationship of the *ARF* as a function of area and duration. The parameters $b_1$, $b_2$, and $b_3$ will be calibrated by a stepwise procedure of Equations (3),

(4), (6–8). The calibration procedure is explained along with the data-processing and examples in the results section.

## 3. Application and Results

The analyses conducted in this paper apply discrete values of durations and areas. The averaging of radar data in space and time is quite computationally demanding, and it is, therefore, not feasible to apply finer intervals. The following values of durations are applied: 1, 10, 30, 60, 90, 180, 240, 360, 540, 720, 1080, and 1440 min along with 20 intervals of the area ranging from 0.25 km$^2$ (one pixel) to 100 km$^2$ (20 × 20 pixels). This corresponds to 240 combinations of area and duration for each of the 534 storms leading to 6408 individual storm-centred areal reduction factor relationships.

Development procedure:

(1) Applying Equation (1), a correction of the pixel scale error is performed. Results are shown for selected durations in Figure 3 and Table 1. It is evident that the error between rain gauge intensities and radar intensities are significantly larger for short rainfall durations. This is a result of the daily mean-field bias adjustment and leads to a bias factor of 1 for the 1440 min durations (1 day). As shown in Figure 3, there is a considerable scatter between maximum rain gauge intensities and the corresponding radar intensities, which is also explained by the Nash–Sutcliffe Efficiency (NSE)-values in Table 1 and Figure 3. Furthermore, the scatter is larger for the shorter durations indicating high uncertainties. However, as the study aims for a mean pixel scale error, the dispersion of the pixel scale error is not considered any further.

(2) For each duration, each individual *ARF*-relationship (Equation (3)) is fitted to Equation (4), providing a unique value of the correlation length, λ, for each duration and storm. Examples of the derived individual storm *ARFs* (Equation (3)) are shown in Figures 4 and 5 in grey.

(3) The correlation lengths, λ are fitted (Equation (6)) as a function of duration (Figure 6). From Figure 6, it is evident that there is a large variability from storm to storm, but that the mean fit well to the power function with r$^2$ of 0.98. It shows that the power-law function in Equation (6) can be further used to derive a relationship of the storm-centred *ARF* as a function of area and duration. In addition to the mean relationship, the uncertainty corresponding to mean plus/minus one standard deviation (assuming a Gaussian distribution) is investigated. This uncertainty will provide insight into the variability from storm to storm.

(4) Applying the obtained function of correlation length and duration, each storm is re-fitted by the relationship in Equation (7) to derive an *ARF* function. Examples of this fit are shown in Figures 4 and 5 for durations of 60 and 360 min, respectively. Comparing with the mean *ARF* functions, the fitted relationships show a slight overestimation for the small areas and correspondingly an underestimation for large areas. For some durations, the opposite case occurs (not shown). This uncertainty is a trade-off of fitting a fixed parameter relationship to all durations.

(5) As a result, the three generic parameters of Equation (8) are calibrated and presented as the mean *ARF*-relationships. Figure 7 shows the obtained relationship for selected durations. Corresponding parameter values are presented in Table 2.

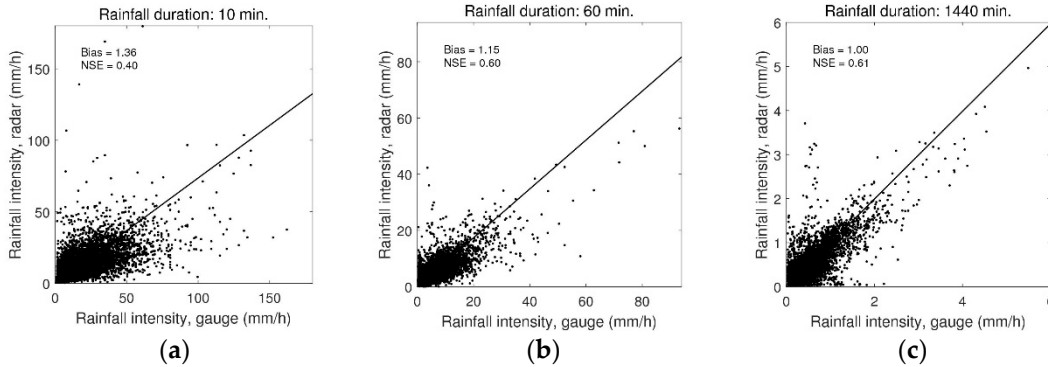

**Figure 3.** Example of estimation of bias between radar and rain gauge intensities at rainfall durations of 10 (**a**), 60 (**b**) and 1440 (**c**) min for the $500 \times 500$ m$^2$ resolution.

**Table 1.** Bias between radar and rain gauge intensities at different rainfall durations for the pixel size of $500 \times 500$ m$^2$.

| Duration, $d$ (min) | 1 | 10 | 30 | 60 | 180 | 360 | 720 | 1440 |
|---|---|---|---|---|---|---|---|---|
| Bias, $B$ (-) | 1.63 | 1.36 | 1.21 | 1.15 | 1.07 | 1.04 | 1.03 | 1.00 |
| Nash–Sutcliffe Efficiency, NSE (-) | 0.21 | 0.40 | 0.52 | 0.60 | 0.63 | 0.62 | 0.62 | 0.61 |
| Root mean square error, RMSE (mm/h) | 22.47 | 9.82 | 4.61 | 2.65 | 1.12 | 0.66 | 0.38 | 0.21 |

It is evident that the variability from storm to storm of the storm-centred *ARF* functions is large compared to mean and fitted *ARF* curves in Figures 4 and 5. This large variability of the storm-centred approach is also shown by Wright et al. [18] for an American radar rainfall dataset in North Carolina. The variability indicates that a variety of different storms is represented in the dataset, i.e., both short convective thunderstorms with a small spatial extent and more widespread rainfall which occurs during stratiform conditions.

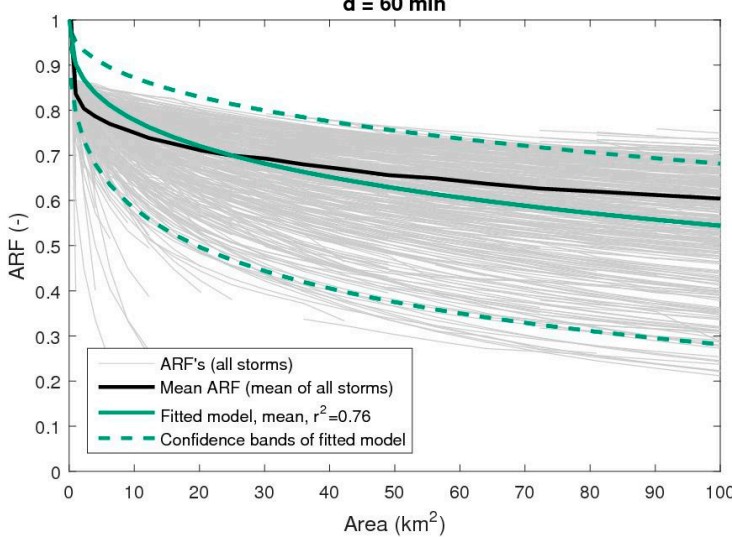

**Figure 4.** Areal reduction factors (*ARF*) for durations of 60 min. Dashed lines are the confidence limits (mean +/− std. dev.) of the fitted *ARF*.

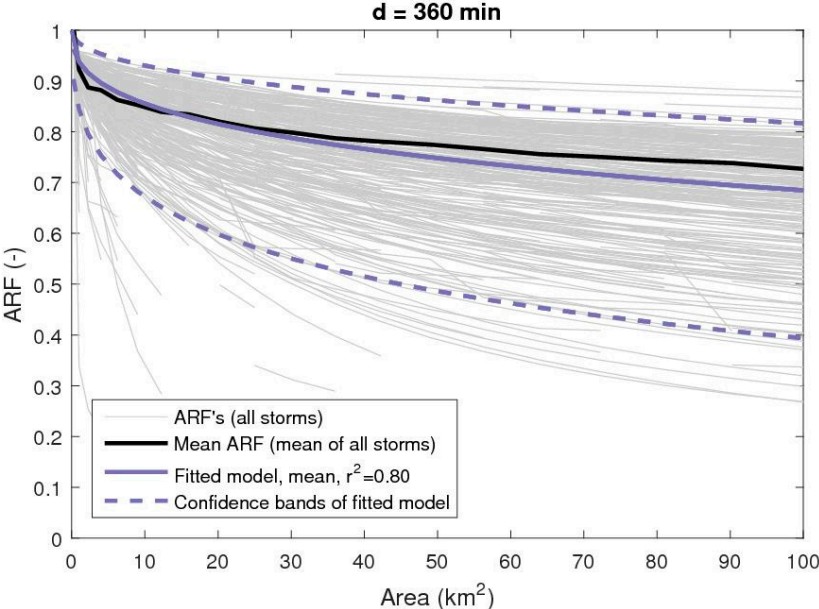

**Figure 5.** *ARF* for durations of 360 min. Dashed lines are the confidence limits (mean +/− std. dev.) of the fitted *ARF*.

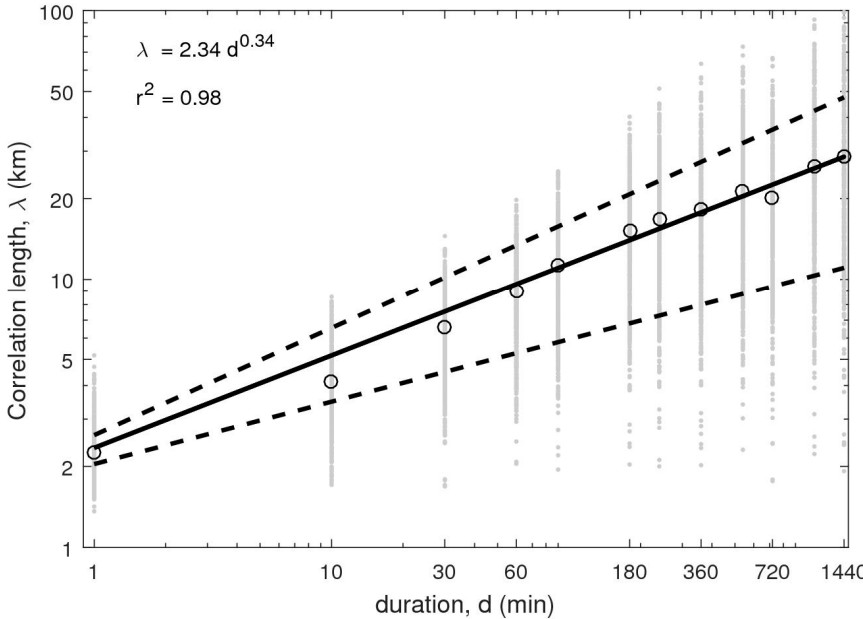

**Figure 6.** Fitted power-function (Equation (6)) between correlation length and rainfall duration in solid black with confidence limits (mean +/− std. dev.) in dashed lines. Grey dots indicate values for individual storms, and black circles indicate mean values for each duration.

**Table 2.** Calibrated values of the three parameters of Equation (8) for mean and mean +/− std. dev.

|  | $b_1$ | $b_2$ | $b_3$ |
| --- | --- | --- | --- |
| *mean* | 0.31 | 0.38 | 0.26 |
| *mean − 1 × std. dev.* | 0.21 | 0.45 | 0.36 |
| *mean + 1 × std. dev.* | 0.47 | 0.37 | 0.17 |

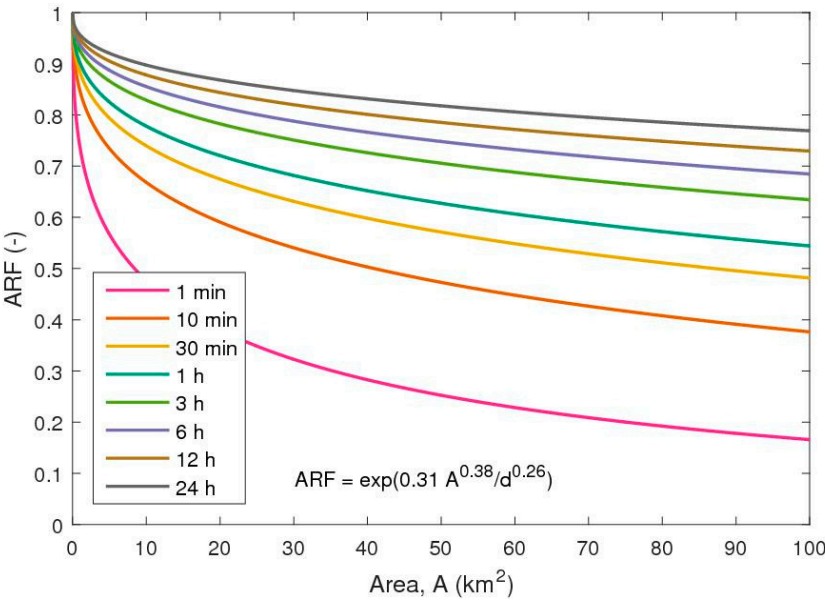

**Figure 7.** Mean areal reduction factors as a function of area and duration.

## 4. Discussion

In order to apply developed *ARF* relationships in design of urban drainage systems, one approach would be to link the extreme event statistics (design rainfall) to the ARF-relationship, e.g., by introducing an *ARF* which is also dependent on the average return period of the design rainfall as presented in Lombardo et al. [16] and Overeem et al. [17]. Because a discontinuous and incomplete radar dataset over the observation period of 15 years is used (as presented in the data section), it is not possible to derive the return periods for rainfall intensities. Likewise, it is not possible to directly transfer the *ARF* relationships to design rainfall. This is a limitation of this study. In comparison with other studies, the rainfall database contains a large quantity of data over a large domain, and a significantly better description of the spatial rainfall variability, than if rain station/gauge records were used alone. This means that the *ARF* relationships are based on more data than previous studies (see Section 4.1). This enhances the validity of the study.

If average return periods of design rainfall were linked to *ARF* relationships, it would require the application of an area-fixed approach instead of the storm-centred approach. Due to the reasons indicated above, this is a subject of further investigations. Following the argument of Wright et al. [18], there are also difficulties using the area-fixed approach, in terms of representing the "true" properties of more extreme events (i.e., longer return periods). In their conclusions, this might lead to larger *ARF* values (compared to the storm-centred approach), thus to overestimation of design rainfall and again to oversizing of urban drainage systems to be designed. Whether the area-fixed approach would result in different *ARF* relationships are in this case, a subject of further investigation. This study, however, limits to conclude that there is a quantifiable uncertainty of the *ARF* relationships from storm to storm, which is important to consider in any design phase.

The dependence of the return period is not supported by the findings of Wright et al. [18] who argue that it is an artefact of the chosen method. Furthermore, it is questionable if the maximum point rainfall would have the same return period as the maximum areal rainfall since they might originate from very different weather types (i.e., convective vs. stratiform storms). This is thus an argument in favour of using the storm-centred approach where the maximum areal and maximum point rainfall occur at the same time. Whether or not the return period dependence is present or whether it should be considered in urban drainage design, is thus an open question. Again, this study acknowledges the variability in the *ARF* from storm to storm and suggests that mean values *ARFs* plus/minus one or two standard deviations are used according to the required safety in the design.

### 4.1. Comparison with Previous Studies

To discuss the generalisation of results, an example of the variability of *ARFs* found in the previous studies are given below. This highlights how the *ARFs* vary depending on the method, data quantity and quality, national guidelines or experience, choice of spatial interpolation method (for rain gauge data), location, climate, etc. Values relevant within an urban hydrological context, i.e., *ARFs* for 1 h durations with areas of 10, 50, and 100 km$^2$ are examined in Figure 8. Few authors [11,16,17,25] have reported *ARFs* for sub-hourly durations and areas less than 100 km$^2$. This comparison is not exhaustive, but it gives an indication of differences between *ARF* values using the area-fixed or storm-centred approach as well as differences using rain gauge and radar data.

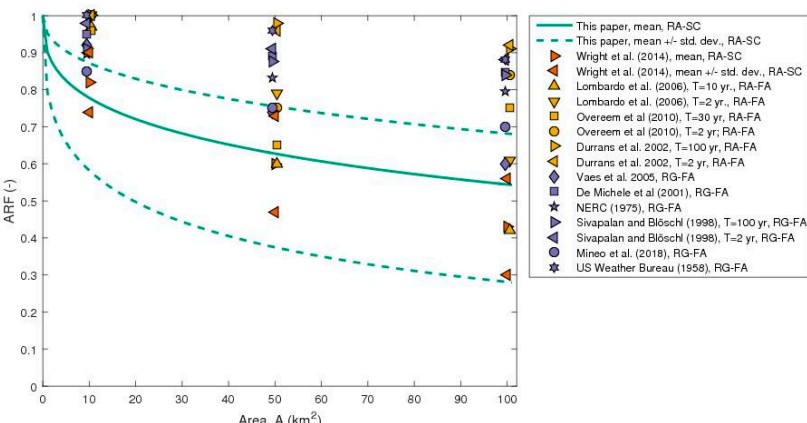

**Figure 8.** Derived *ARF* relationship for 60 min durations compared to previous studies for areas of 10, 50 and 100 km$^2$ estimated using fixed-area (FA) and storm-centred (SC) approaches applying both rain gauge (RG) and radar data (RA).

From Figure 8, there is a tendency for *ARFs* derived from rain gauge data to show higher values than the radar data counterparts. Furthermore, there is a tendency for storm-centred *ARFs* derived in this paper and in Wright et al. [18] to be smaller than the ones found using area-fixed approaches. Smaller values of the storm-centred approach in comparison to the area-fixed have also been stated in Sivapalan and Blöschl [5]. Many of the fixed-area *ARFs* have been developed for much larger spatial scales (e.g., up to 20,000 km$^2$, [15]) and for rainfall durations of 1 day or more. Calibrated relationships which cover these large rural scales might potentially be less precise on smaller scales relevant in urban hydrology. It is acknowledged that the *ARFs* derived in this paper are smaller than the ones reported in the majority of previous studies. Mean values and standard deviations are, however, comparable to the ones reported by Wright et al. [18], which implies that the storm-centred approach will provide larger areal reductions, thus smaller *ARFs*, compared to the fixed-area counterparts. From the comparison, it is not possible to determine whether *ARFs* obtained from rain gauge data is different from the ones based on radar data.

### 4.2. Implementation in Urban Drainage Design

As stated in the introduction, it has not previously been a part of the code of practice in Denmark to account for the spatial variability of rainfall in the design of urban drainage systems. Thus an *ARF* equal to 1 has been applied independently of design duration and catchment area. Introduction of the *ARF* will lead to a reduction in the design rainfall intensities for increasing catchment sizes. Following Allen and DeGaetano [15], Lombardo et al. [16] and Overeem et al. [17], increasing the average return period will also give reason to decrease the *ARF* and thus, the design rainfall intensity. Relating this to the mean and confidence intervals presented in Figures 4 and 5 will entail that the upper confidence limit (*mean + 1 × std. dev.*) will correspond to lower return periods, and the lower confidence limit (*mean − 1 × std. dev.*) will correspond to higher return periods. Since design rainfall

intensities will increase as a function of increasing return period, the *ARF* would counteract and lead to smaller values; thus, smaller design rainfall intensities for increasing return periods. It is therefore argued, that it would be on the safe side to use the mean *ARF*, but still, a significant reduction of the design rainfall compared to neglecting the areal component. It is recommended that the best option is to use the mean *ARF* as the most probable value with lower and upper confidence limits of +/− 1 standard deviation.

Whilst most other authors have focused on developing areal reduction factors for large hydrological scales, short durations (<1 h) and small catchments (<10 km$^2$) are of importance in urban hydrology. An example relevant to urban scales is presented in the following, along with an evaluation of the impact of the implementation of an *ARF*.

For a catchment of 10 km$^2$ with a time of concentration corresponding to 60 min to the outlet, the developed mean *ARF* is 0.78 with 0.58 and 0.87 for the lower and upper confidence limits (+/- the standard deviation), respectively. This corresponds to a reduction of the design rainfall on average of 25% ranging from 13% to 46%. Using the mean *ARF*, the design rainfall for Danish conditions with a return period of 10 years [39] can be adjusted as presented in Figure 9a for *IDF* curves and Figure 9b for design rainfall of the Chicago design storm (CDS) type [40]. Compared to the current design practice where no areal reduction is implemented; this will reduce the design intensities and thus lead to smaller designs.

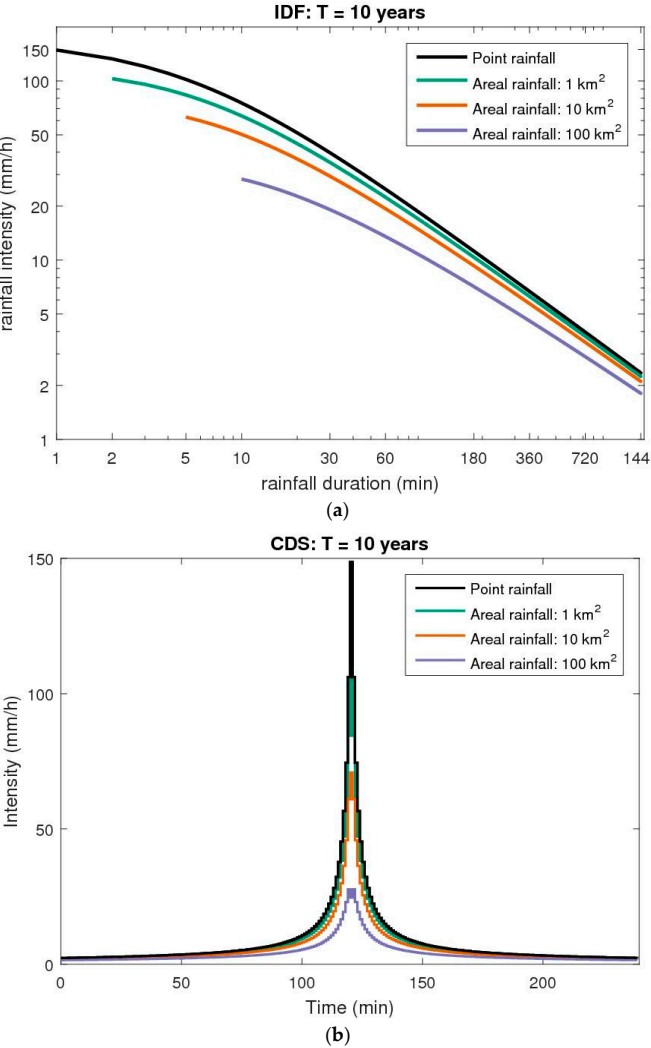

**Figure 9.** Example of intensity–duration–frequency curves (**a**) and Chicago design storms (**b**) with a return period of 10 years [2] adjusted with the developed mean areal reduction factors.

## 5. Conclusions

In this paper, storm-centred areal reduction factor relationships were developed from a radar rainfall dataset covering a period of 15 years. A novel and generic *3-parameter* relationship describing the areal reduction factors as a function of rainfall duration and area has been derived and calibrated against 6408 individual combinations of storm, area, and duration. The results showed that there is a large variability in the developed *ARFs* for individual storms; however, using mean values of the developed *ARF*, combined with confidence limits, provides a reliable method to adjust rainfall for urban drainage design purposes. Impacts of using the storm- centred approach, compared to the more traditional area-fixed approach linked to average return periods of design rainfall, is however still subject to further investigations.

The derived areal reduction factor relationships highlight the importance of accounting for areal rainfall variability in the design of urban drainage systems with catchments as small as 10 km$^2$ with short concentration times (thus short rainfall design durations), to prevent unnecessary oversizing of designs.

**Author Contributions:** Conceptualisation, methodology, formal analysis, and writing—original draft preparation, S.T. Software, writing—review and editing, and validation, J.E.N. Project administration writing—review and editing, and validation, M.R.R.

**Funding:** This research was partially funded by Aarhus Water and VUDP (Vandsektorens Udviklings-og Demonstrations program, Denmark, project ID 1162.2017).

**Acknowledgments:** The authors would like to thank the Danish Meteorological Institute for the use of radar data and the Danish Wastewater Committee under the Society of Danish Engineers for rain station/gauge data.

**Conflicts of Interest:** The authors declare no conflict of interest.

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
