# Peer review of "Estimation of Storm-Centred Areal Reduction Factors from Radar Rainfall for Design in Urban Hydrology"

_water, doi:10.3390/w11061120_

Round 1

Reviewer 1 Report

In this study authors proposed a method for estimation of spatial variability of rainfall in small urban hydrological systems using areal reduction factors. These factors were established from a large radar rainfall dataset based on the storm-centred approach. The results indicate that there is a very large variability in the estimated areal reduction factors for individual storms. Therefore, the authors suggest that in design of small urban drainage systems to use the mean values of areal reduction factors, calculated using the expression established in this study as a function of rainfall duration and area.

The manuscript is interesting and well structured. I suggests the article can be accepted after minor revisions. The following are issues that can be improved:

It is advisable to explain in a more precise and unambiguous way which specifically elements of the research are the novelty in relation to other research in this area. This additional briefly explanation should be added at least in the Introduction.

Please briefly explain in chapter 2.1 why the radar rainfall data generated in a 1 km altitude was chosen for the analysis as well as why the data from the radar product of pseudo-CAPPI was used (it was due to the availability of such data or was justified by scientific reasons?; data generated for example in a 0.7 km altitude was less useful?).

Author Response

Reviewer 1

Comments and Suggestions for Authors

In this study authors proposed a method for estimation of spatial variability of rainfall in small urban hydrological systems using areal reduction factors. These factors were established from a large radar rainfall dataset based on the storm-centred approach. The results indicate that there is a very large variability in the estimated areal reduction factors for individual storms. Therefore, the authors suggest that in design of small urban drainage systems to use the mean values of areal reduction factors, calculated using the expression established in this study as a function of rainfall duration and area.

The manuscript is interesting and well structured. I suggests the article can be accepted after minor revisions. The following are issues that can be improved:

REPLY: Thank you for the review. I have replied to your comments below.

 It is advisable to explain in a more precise and unambiguous way which specifically elements of the research are the novelty in relation to other research in this area. This additional briefly explanation should be added at least in the Introduction.

REPLY: we added two sentences on novelty in the introduction.

Please briefly explain in chapter 2.1 why the radar rainfall data generated in a 1 km altitude was chosen for the analysis as well as why the data from the radar product of pseudo-CAPPI was used (it was due to the availability of such data or was justified by scientific reasons?; data generated for example in a 0.7 km altitude was less useful?).

REPLY: we added this to explain in more detail: “This product is selected due to consistency in data having a fixed elevation as function of range and significantly less errors and clutter compared to lower altitude pseudo CAPPI products.”

Reviewer 2 Report

In overall, the manuscript presents a good idea of bias correction in spatial and temporal rainfall distribution which is essential for small catchment flood estimation. However, the manuscript is written in very poor English. This leads to misunderstanding and confusion. The manuscript must be rewritten to meet the standard of publication. I am between rejection and major change. If the authors are able to rewrite the manuscript in academic English, using hydrological terminologies, it may be qualified for publication.

Some comments are below:

About content:

-        Equation 1 (p.1,l119): - (1) What is G? There are two ways to express maximum rainfall intensity: using Max function and define G or using imax and define what imax is. (2) What is “daily mean-field bias”. What equation has been used to calculate this variable? Please clarify, readers would not go to citation [30] to find what “daily mean-field bias” is. (3) if iR is “daily”, is it necessary to include d in the denominator?

-        Are Figure 2 and Table 1 (p.4, l126-130) the authors’ work or someone’s else work. Please cite it if it is necessary. 

-        Section “Methodology” is very confused:

          Area A (p.5,l140) cannot be 0. It approaches 0 (A → 0). Please change it.

          Equation 4 is not related to other equations. I do not think that it is necessary here.

          Equation 5 has little meaning since the authors do not use it further. Exclude it.

-        The statement “By grouping the fitted values of for each storm by the duration, we found that a power-function approximation of with coefficients a1 and a2 as a function of d provides acceptable results (see discussion below):” (p5, L170-171) needs clarification. It is not clear where this relationship comes from. In what section this statement has been discussed or proved?

-        Figure 5 (p7, l210): Using confidence limits of 90% (5% each side) instead of (mean +/- std) will be more appropriate.

About writing: The manuscript writing is not academic. It is more like verbal than written style. There are too many vague words and phrases; Many sentences confuse me. Terminologies are used in the wrong way. For example (Please be advised that I do not list all of them herein),

    - The abstract has many vague words and phrases: “… is most often assumed spatially homogeneous over a given catchment ” (P.1, L.11); “By extending methods from rural hydrology,..” (P.1, L.13-14); “… 0 to 100 km2” (P.1,I.16) (Area of 0 does not make sense); “.. a novel general expression …” (P.1,L.17) (It should be “an empirical equation” or “relationship”). Other examples of vague phrases are “smaller values thus smaller design rainfall intensities” (P.10, L.283), “certain safety” (P.10,) “This large variability…” (P.6, L.200),

 - Terminology: The terminology “empirical” is used in the wrong context. There are no empirical points. I guess the authors mean the recorded or observed points. The terminology “general expression” should be “empirical equation” or “empirical coefficients” (depending on context) which is the relationship derived from analysis of records. P.3, L.105 states “In traditional rain gauge-based investigations …”. “Traditional methods” or “traditional approach” are common sentences. There is no such thing “traditional investigation”.

- Section “Results and discussion”. I am totally confused with the section from “We finish this discussion by giving an example of variability of ARFs found in the cited literature ….” to the end of the discussion (before section 3.1). The authors show differences in results between the existing investigation with the previous studies. What is the conclusion from this comparison? If “We acknowledge that the ARFs derived in this paper are indeed smaller than the equivalent reported in the majority of literature”, which one is more reliable and should be used? The existing results or others?   

- “Again, we acknowledge the great variability in the derived empirical ARF and suggest applying a mean value for design in Denmark.” (P.9, L.249-250). I do not understand what the authors want to claim.

- It will be more appropriate if the manuscript uses the third party in all sentences instead of using “We”.  

- There are many unnecessary citations in section of results and analysis.

Author Response

Reviewer 2

Comments and Suggestions for Authors

In overall, the manuscript presents a good idea of bias correction in spatial and temporal rainfall distribution which is essential for small catchment flood estimation. However, the manuscript is written in very poor English. This leads to misunderstanding and confusion. The manuscript must be rewritten to meet the standard of publication. I am between rejection and major change. If the authors are able to rewrite the manuscript in academic English, using hydrological terminologies, it may be qualified for publication.

REPLY: Dear reviewer 2. Thank you for your review. I think you have pointed-out some weaknesses of the paper, which are indeed relevant. Throughout out the paper, I have added and corrected according to your comments, and in my view it has significantly improved the quality of the paper. It seems that you find my writing problematic. I have deliberately chosen a more modern academic style of writing scientific papers, where I use “we” instead of a third-person approach.

Places where the written English has been too misleading I have changed, but I will let it be up to the editor to decide whether the academic English in general should be reconsidered. The manuscript has been corrected by a native speaker.

Below, I have explained the changes made in the manuscript and replied on the reviewers comments:

Some comments are below:

About content:

-        Equation 1 (p.1,l119): - (1) What is G? There are two ways to express maximum rainfall intensity: using Max function and define G or using imax and define what imax is. (2) What is “daily mean-field bias”. What equation has been used to calculate this variable? Please clarify, readers would not go to citation [30] to find what “daily mean-field bias” is. (3) if iR is “daily”, is it necessary to include d in the denominator?

REPLY: I have defined G and R and explained the procedure in more detail and explained the max function. d is for duration and not daily. I have added more on the mean-field bias in section 2.1 to clarify this

-        Are Figure 2 and Table 1 (p.4, l126-130) the authors’ work or someone’s else work. Please cite it if it is necessary.

REPLY. This is indeed our own work and it is based on the same dataset that the use throughout the paper.

-        Section “Methodology” is very confused:

Area A (p.5,l140) cannot be 0. It approaches 0 (A → 0). Please change it.

REPLY: Changed

Equation 4 is not related to other equations. I do not think that it is necessary here.

REPLY: It is indeed important in order to develop eq. 5

Equation 5 has little meaning since the authors do not use it further. Exclude it.

REPLY: We do use it. I should be more clear after explaining the procedure in more detail as requested by one of the other reviewers. The  procedure is added in the end of section 2.3

-        The statement “By grouping the fitted values of for each storm by the duration, we found that a power-function approximation of with coefficients a1 and a2 as a function of d provides acceptable results (see discussion below):” (p5, L170-171) needs clarification. It is not clear where this relationship comes from. In what section this statement has been discussed or proved?

REPLY: from the added explanations in section 2.3 and the procedure in the results section it should be more clear how the procedure is developed

-        Figure 5 (p7, l210): Using confidence limits of 90% (5% each side) instead of (mean +/- std) will be more appropriate.

REPLY: It is very deliberately that we show the confidence limits corresponding to confidence limits of 68% (~16 % on each side) since we use +/-  one times the standard deviation throughout the paper

About writing: The manuscript writing is not academic. It is more like verbal than written style. There are too many vague words and phrases; Many sentences confuse me. Terminologies are used in the wrong way. For example (Please be advised that I do not list all of them herein),

    - The abstract has many vague words and phrases: “… is most often assumed spatially homogeneous over a given catchment ” (P.1, L.11); “By extending methods from rural hydrology,..” (P.1, L.13-14); “… 0 to 100 km2” (P.1,I.16) (Area of 0 does not make sense); “.. a novel general expression …” (P.1,L.17) (It should be “an empirical equation” or “relationship”). Other examples of vague phrases are “smaller values thus smaller design rainfall intensities” (P.10, L.283), “certain safety” (P.10,) “This large variability…” (P.6, L.200),

REPLY: The words and phrases are reformulated to be more precise throughout the paper. I have deliberately chosen a more modern academic style of writing scientific papers, where I use “we” instead of a third-person approach. I think it makes the paper flow better and is merely a matter of taste than whether it is correct academic English. I will let that up to the editor to decide.

 - Terminology: The terminology “empirical” is used in the wrong context. There are no empirical points. I guess the authors mean the recorded or observed points. The terminology “general expression” should be “empirical equation” or “empirical coefficients” (depending on context) which is the relationship derived from analysis of records. P.3, L.105 states “In traditional rain gauge-based investigations …”. “Traditional methods” or “traditional approach” are common sentences. There is no such thing “traditional investigation”.

Reply: Good points. I have edited throughout the paper.  The “general expression” has been changed throughout the paper to empirical relationship.

- Section “Results and discussion”. I am totally confused with the section from “We finish this discussion by giving an example of variability of ARFs found in the cited literature ….” to the end of the discussion (before section 3.1). The authors show differences in results between the existing investigation with the previous studies. What is the conclusion from this comparison? If “We acknowledge that the ARFs derived in this paper are indeed smaller than the equivalent reported in the majority of literature”, which one is more reliable and should be used? The existing results or others?  

REPLY: we added as separate discussion section with the comparison to literature values and added more text to generalize and conclude on the comparison. I hope this is more clear

- “Again, we acknowledge the great variability in the derived empirical ARF and suggest applying a mean value for design in Denmark.” (P.9, L.249-250). I do not understand what the authors want to claim.

REPLY: This is changed to : “Again, we acknowledge the great variability in the derived empirical ARF from storm to storm. We suggest that depending on the design situation and required safety in design, that mean values ARF’s plus/minus one or two times the standard deviations are used”

- It will be more appropriate if the manuscript uses the third party in all sentences instead of using “We”. 

Reply: see reply on this above

- There are many unnecessary citations in section of results and analysis.

Reply: I think the citations are important in order to validate the results and to compare to results of other authors. Some of the text has been moved to the discussion section

Reviewer 3 Report

The Authors present a further development of ARF methods, to be applied in urban environments. The paper is well structured and well written. The introduction is complete, the method is scientifically sound and the results are well presented. The only major issue I have after reading the paper is that the method section is a bit convoluted (please see some specific comments below). I feel that some further explanation is needed on how exactly the method is applied. I suggest adding another figure that will go along with the method description and will illustrate the steps needed to compute the ARF using the methods described in the paper. I am suggesting the paper for a major revision before accepting it to Water (although I have full confidence that the paper will be of interest for the reader of Water after the revisions will be made).

Specific comments

[16] Cannot be zero. Maybe 0.1?

[36] The sentence is not completely clear, please revise.

[48-50] Whom? I suggest adding two or three references to support this sentence.

[59] "on short rainfall duration" - Minute to hour scales?

[64] Delete the word "with"

[66-67] Repetitive of line 59. There are several repetitions in the text, please revise it carefully.

[109] "true"

[118] I suggest not mixing terms here - subpixel variability refer to the distribution of multiple values within a given radar pixel, here you compute the bias, which can be computed using several gauges within a radar pixel, but will not represent the true natural variability.

[122] I agree with the rationale of computing the bias the way you are suggesting, but I wonder if by using all rain gauges in the radar domain, instead of dividing the radar domain to several climatological regions, you are not artificially increasing the computed bias.

[126] 1440. I do not understand how the bias at the daily scale can be equal to exactly 1. Let's assume that you have a radar pixel that covers 3 rain gauges - maybe one of them (after bias correction) will be identical to the estimate of the radar, but the chances that the other two gauges will also have a bias of 1 is very low... In addition, examining the right plot at Fig. 2 it seems that the bias should deviate from being perfect 1.

[eq 3] Looking at equation 3, I am now more concern about the fact that the bias is calculated over the entire radar domain. Please consider computing "regional" bias for several locations and compare with the bias you computed over the entire domain to explore the sensitivity of the bias to the ARF.

[eq 9] I agree that Eq. 9 is general, but also have a relatively high degree of freedom as you have 4 parameters to fit (a1, c1, c2 and a2). Or do you fit only three parameters (b1, b2 and b3)? This is not clear to me. Also - how the parameters are fitted and how the bias that was computed before (Eq. 1) is related to Eq. 9? Please clarify (maybe add a figure to explain the chain of actions needed by the user).

[182] For how many locations? From the text, if reads like you performed this analysis for all radar pixels...

[Fig. 8] Looking at figure 8 raise the question of how general are your results. Consider adding a sentence discussing this.

Author Response

Reviewer 3

Comments and Suggestions for Authors

The Authors present a further development of ARF methods, to be applied in urban environments. The paper is well structured and well written. The introduction is complete, the method is scientifically sound and the results are well presented. The only major issue I have after reading the paper is that the method section is a bit convoluted (please see some specific comments below). I feel that some further explanation is needed on how exactly the method is applied. I suggest adding another figure that will go along with the method description and will illustrate the steps needed to compute the ARF using the methods described in the paper. I am suggesting the paper for a major revision before accepting it to Water (although I have full confidence that the paper will be of interest for the reader of Water after the revisions will be made).

Reply: Thank you for the review. I think that the comments has helped to improve the quality of the manuscript. Please find reply to the comments below.

Specific comments

[16] Cannot be zero. Maybe 0.1?

Reply: corrected

[36] The sentence is not completely clear, please revise.

Reply: revised

[48-50] Whom? I suggest adding two or three references to support this sentence.

Reply: sentence is changes and references are added

[59] "on short rainfall duration" - Minute to hour scales?

Reply: revised

[64] Delete the word "with"

Reply: revised

[66-67] Repetitive of line 59. There are several repetitions in the text, please revise it carefully.

Reply: repititions are deleted

[109] "true"

Reply: revised

[118] I suggest not mixing terms here - subpixel variability refer to the distribution of multiple values within a given radar pixel, here you compute the bias, which can be computed using several gauges within a radar pixel, but will not represent the true natural variability.

REPLY: Good point - I have changed the term to “pixel scale error”.

[122] I agree with the rationale of computing the bias the way you are suggesting, but I wonder if by using all rain gauges in the radar domain, instead of dividing the radar domain to several climatological regions, you are not artificially increasing the computed bias.

[126] 1440. I do not understand how the bias at the daily scale can be equal to exactly 1. Let's assume that you have a radar pixel that covers 3 rain gauges - maybe one of them (after bias correction) will be identical to the estimate of the radar, but the chances that the other two gauges will also have a bias of 1 is very low... In addition, examining the right plot at Fig. 2 it seems that the bias should deviate from being perfect 1.

REPLY: Some more explanation on the bias adjustment have been added in section 2.1 to clarify that there is only one gauge per pixel. Also NSE values have been added to table 1 and figure 2 in order to show the variability in the bias estimates. I have added more text to explain.

[eq 3] Looking at equation 3, I am now more concern about the fact that the bias is calculated over the entire radar domain. Please consider computing "regional" bias for several locations and compare with the bias you computed over the entire domain to explore the sensitivity of the bias to the ARF.

Reply: Using the MFB does not provide any possibility to use a regional bias. This is also argued in Thorndahl et al (2014) which provides the  data foundation for this manuscript. I have added a sentence to explain this:

[eq 9] I agree that Eq. 9 is general, but also have a relatively high degree of freedom as you have 4 parameters to fit (a1, c1, c2 and a2). Or do you fit only three parameters (b1, b2 and b3)? This is not clear to me. Also - how the parameters are fitted and how the bias that was computed before (Eq. 1) is related to Eq. 9? Please clarify (maybe add a figure to explain the chain of actions needed by the user).

Reply In the result section I have added more on the development procedure. It should be more clear now that we first need to fit c and a parameters before ending up with the 3-parameter equation (9)

[182] For how many locations? From the text, if reads like you performed this analysis for all radar pixels...

Reply: We you the pixels with maximum intensity for each duration for each storm. This should also be more clear form the added procedure. It is also explained below eq. 2

[Fig. 8] Looking at figure 8 raise the question of how general are your results. Consider adding a sentence discussing this.

REPLY: we added as separate discussion section with the comparison to literature values and added more text to generalize and conclude on the comparison.

Round 2

Reviewer 2 Report

Please see attached file for comments

Author Response

Comments on manuscript “Estimation of storm-centred areal reduction factors from radar rainfall for design in urban hydrology”

The new version is not much improved compared with the previous version. There is no novelty in approaches and methodology used. However, I accept for major change because the study entirely present a case study.

The written style of the manuscript does not meet requirements for publication. I recommend that the manuscript should be submitted to professionally editing before submission or read by a native English-speaking colleague.

REPLY: Dear Reviewer 2. Thank you very much for your very profound review of our paper. I agree with you that there is not much novelty in the methology, but I think the urban application as well as the developed relationships (the 3-parameter relationship of the ARF as function of area and duration) is indeed novel.

I have changes a lot of the formulations according to your comments and very qualified suggestions to rephrasing. I think the general meaning of the papers is much clearer now. According to your suggestions I have also changes the written style so that it hopefully now will meet the requirements of Water. I have for example changed to third person throughout the paper.

Please find more detailed replies below

About the content:

1.     Equation 1 (p.1,l119): if  iR is is not daily rainfall intensity, it should be clarified that iR is the rainfall intensity over duration d which calculated from daily mean-field bias …”

REPLY: revised accordingly

2.    Are Figure 2 and Table 1 (p.4, l126-130) the authors’ work or someone’s else work? If they are the authors’ work and have been published somewhere else, citation is needed.

If they are results of the current study, it should not be included in methodology section. It must be described in section of results.

REPLY: The data is indeed the authors own work. Parts of section 2.2 have been moved to section 3 and some additional text has been added

3.    Equation 4 is not related to other equations. I do not think that it is necessary here. (REPLY: It is indeed important in order to develop eq. 5)

Equation 4 was used in the previous studies to derive equation 5, not in the current study. In this study, equation 5 is referenced from the previous study and has been used for further development.  This also is cited. This is sufficient and no need to backdate to eq.4. If you want to include equation 4, it is necessary to explain how  links with other variables and has been used in further analysis. I have not seen  used in the next sections.

REPLY: Eq. 4 has been deleted and numbering has changed accordingly

4.    The results are not quite convincing as Nash-Sutcliffe coefficient values are too low (see table 1). However, I aware that the rainfall data are highly variable

REPLY: I agree that NSE values are indeed low for the small durations due to a significant scatter. This does not, however, affect the estimation of the mean bias that we use in this study. The small NSE values are explained in the first point of the development procedure.

As suggested by one of the other reviewers tne RMSE-values are added in table 1

About writing:

5.    Wording is inaccurate, for example:  “empirical”  “Fit” and “fitting”:

-          L18, P1. “a novel empirical relationship”: “novel” should not be. “novel” and “empirical” can not be put in one phrase in this case as their meanings are opposite.  

REPLY: empirical is deleted

-          As I mentioned before, the word “empirical” is overused and sometimes is used in wrong context. I suggest the authors read more about term of “empirical” and therefore can use this word efficiently. For example, “The empirical areal reduction factor (ARFemp)” (L.163 ,P.5) should be “The areal reduction factor”.

REPLY: empirical is deleted throughout

-          “Where c1 and c2 are fitted coefficients” (L202, P.6) should be “where c1 and c2 are coefficients fitted/calibrated from record data”. “Fit/fitting” is preferred in statistical analysis such as “fit a distribution”, meanwhile “Calibrate/calibration” is commonly used in modelling approach. The use of “calibration” or “fit” will depend on context in this manuscript.

REPLY: Good point. “fit” has been changed to “calibrate” when it describes the ARF model. In cases where mathematical functions are paramerized, “fit” is maintained.

-          “Comparsion with litlerature values”: should it be “Comparison with previous studies” ?

-          “the catalogue of rainfall days”: should it be “rainfall database” or “rainfall dataset”?

REPLY: revised

6.    The third-person use:

It is important in every research that authors have an unbiased view about the proposed investigation issues. The use of the first-person “We” is preferred when the authors want to emphasize a personal opinion about the approach/method/results used/applied in a particular situation. Meanwhile, the use of the third-person structure will highlight that there are unbiased conclusions drawn from the investigation and these methods/approaches/conclusions can be successfully applied/resulted for all other case studies. 

Please see publications in MDPI to select the right form.

REPLY: I do not necessarily agree with you here, but I respect that a journal can have specific preferences. I therefore changed to third person format throughout the paper

7.    Poor structure, for example:

The individual storms are defined from the catalogue[1] of rainfall days as presented in the data section. We estimate one maximum rainfall intensity per duration per area per rainfall day within the study domain and define this as a storm[2]. We thus neglect[3] the possibility for selecting multiple peaks within the same area and duration in the domain within[4] the same day” (L175 – 178, P.5).

Please see one of possible options to present your idea: “The individual storms are defined from the rainfall dataset presented in the data section. For each storm, the maximum rainfall intensity for duration d is selected. Thus, this will limit the possibility of selecting multiple peaks for a given domain.”

REPLY: catalogue is revised to database. Wording is changed and definition of storm is explained in more detail:

The individual storms are defined from the rainfall database as follows: One maximum rainfall intensity per duration, area, and rainfall day within the study domain is found. In this study, this defines as a storm, knowing that other storm definitions exist in the literature. This storm definition allows for multiple locations of the pixel with the highest rainfall intensity, depending on duration and area within a rainfall day. It is a key feature of the storm-centered approach that the location is not fixed. A limit of the definition is that only one peak (maximum intensity and location of maximum intensity) per storm can be selected.

“Still with the aim of developing an empirical relationship with few parameters of the ARF as a function of both area and duration, we investigate different relationships  between l and d. By grouping the fitted values of l for each storm by the duration, we found that a power-function approximation of l with coefficients a1 and a2 as function of d provides a good fit of the correlation length as function of the duration (see the obtained function and discussion of results in section 3):” (L.207-211, P.6).

I suggest a possible paraphrase: “The novelty of this study is to develop a relationship between l and d[5]. Preliminary analysis of record values of l for each duration suggests to use a power-function approximation described by the following equation[6]:”

REPLY: Good point. The paraphrase is revised according to your suggestion

The word “relationship” is overused. For example, it has been used 4 times, in all sentences of the below paragraph:

“From this, we can calculate the three general parameters from eq. 9 and present this as the mean empirical relationship of the derived ARF-relationships. Figure 7 shows the obtained relationship for selected durations. Corresponding parameter values for the simplified relationship (eq. 9) are presented in table 2.” (L264-267, P.7).

REPLY: some of the “relationships” are deleted

I shall NOT list all paragraphs which need professional editing. There are many of them. One of the most confused paragraphs are:

Applying the obtained values of a1 and a2, c1 and c2 can be fitted to the mean (or an arbitrary quantile) of the empirical ARF’s for each storm using a non-linear least squares approximation.”

REPLY: This is changed to : Applying the relationship developed in eq. 6, the parameters c1 and c2 (eq. 5) are fitted with non-linear least squares approximation. This means a modification of the initial guess of parameters from eq. 4. With all four parameters calibrated, Eq. 7 can be simplified to the following relationship:

“Because we use a discontinuous and incomplete radar dataset over the observation period of 15 years (as presented in the data section), we omit from estimating return periods for rainfall intensities and for the developed ARF relationships. This said, however, the dataset still represents a substantial quantity of data over a large area and significantly better description of the spatial rainfall variability than if we use rain gauge data alone. It is likely that the dataset includes return periods larger than the 15 years since several very extreme events were observed within the period, e.g. the 2 July 2011 [30] during which [41] reported that return periods up to 2000 years were recorded at the most critical locations. In order to apply developed ARF relationships directly in design of urban drainage systems, it would be relevant to further link the extreme value statistics to the ARF-relationship, e.g. by introducing an ARF which is also dependent on the return period as presented in [16,17]. This would require applying an area-fixed approach instead of the storm-centred approach, but due to the reasons indicated above, this is a subject of further investigations and potentially combination with rain gauge data to extend length of rain series and return periods.” (L325-336, P.10).

I guess that the authors want to address that the limitation of this study is ignoring probability component of rainfall intensity in estimation of ARFs. However, the paragraph needs clarification.

REPLY:  This section has been rewritten in order to clarify the meaning.

8.    Grammar mistakes: (e.g.: [27–29] (L72, P.2); “figure 2 and table 1” (L141, P.4) – please use capital letter for all Figures and Tables, “Where ρ” (L.192, P6); “Where l” (L.196, P.6):  Lowercase letter (where) should be used. Please apply for all cases of variable definitions.

Some more grammar mistakes: “… we omitted from…” “min.” (should be minute?)

REPLY: revised

9.    Wrong citation:

“review by [20]” (L40, P1); “As argued by [18]” L45-46, P1-2; “[18] argue that” (L.52, P.2); “conclusions of [18]” (L59, P2); “detail in [30]” (L84, P.2); “Following [4,5]” (L.193, P.6). Please find and correct all these citations.

It should be: Author name [citation], for example: As argued by Wright et al. [18].

Please see one of the publication in MDPI https://www.mdpi.com/2073-4441/11/5/986 of how to cite previous studies.

REPLY: Citations included in the text are changes to the correct format

10. References are in the wrong format

Please see one of the publication in MDPI https://www.mdpi.com/2073-4441/11/5/986 of how to list references.

REPLY: revised accordingly

[1] “catalogue” should be database

[2] It is common to identify the storm firstly and then the rainfall intensity of this storm. In this way, it avoid to select multiple peaks as a storm will include both storm duration and area covered.

[3] Should be “limit”

[4] Two words “within” in the same sentence

[5] Using this statement will highlight your contribution in knowledge

[6] This section is to desribe the methodology going to use futher in the next section. Approvement of this equation will be confirmed in result and discussion sections. It is not neccessary to approve this equation in this section.

Reviewer 3 Report

I found this version of the manuscript much improved in comparison to the original submission. I have only a single minor comment that the Authors might want to consider (see below), but other than that I recommend the manuscript for publication.

Minor comment

In table 1 the Authors use NSE index. I suggest also adding/replacing this index with RMSE index, which is more common when comparing radar rainfall estimates.

Author Response

Dear Reviewer 3. Thank you for the review. I have added the RMSE values in the table.

best regards

Søren Thorndahl

Round 3

Reviewer 2 Report

The manuscript is improved. However, there are some terminologies used in a wrong context. Please see more comments and suggestions in the attached files. 

The sentences/phrases which I have highlighted need to be improved. They are either not clearly expressed, inaccurate wording or confused. 

The following paragraph is really confused. It needs to rewrite. 

L158-164, P6: "The individual storms are defined from the rainfall database as follows: One maximum rainfall intensity per duration, area, and rainfall day within the study domain is found. In this study, this defines as a storm, knowing that other storm definitions exist in the literature. This storm definition allows for multiple locations of the point with the highest rainfall intensity, depending on duration and area within a rainfall day. It is a key feature of the storm-centered approach that the location is not fixed. A limit of the definition is that only one peak (maximum intensity and location of maximum intensity) per storm can be selected." 

Please correct the word "empirical" in all Figures. 

Author Response

Reply to comments of reviewer 2 are marked in yellow.

The manuscript is improved. However, there are some terminologies used in a wrong context. Please see more comments and suggestions in the attached files.

The sentences/phrases which I have highlighted need to be improved. They are either not clearly expressed, inaccurate wording or confused.

Dear reviewer 2

Thank you very much for all the suggestions and comments to improve the paper. You have done a very detailed, thorough and constrictive revision of the whole paper. I have changed formulations and added throughout the paper according to each of your points.

The following paragraph is really confused. It needs to rewrite.

L158-164, P6: "The individual storms are defined from the rainfall database as follows: One maximum rainfall intensity per duration, area, and rainfall day within the study domain is found. In this study, this defines as a storm, knowing that other storm definitions exist in the literature. This storm definition allows for multiple locations of the point with the highest rainfall intensity, depending on duration and area within a rainfall day. It is a key feature of the storm-centered approach that the location is not fixed. A limit of the definition is that only one peak (maximum intensity and location of maximum intensity) per storm can be selected."

Reply: This section is completely changed  and restructured so it should be more clear how the storms are defined

Please correct the word "empirical" in all Figures.

REPLY: revised

A few other replies

Your point on rain gauge vs. station is acknowledged. I have changed rain gauge to rain station/gauge in the text where it is obvious that at station is addressed. I however like rain station/gauge better than rainfall station/gauge.

At some point you questioned he use of return period and suggested recurrence interval instead. Since I use the term return period quite a lot during the paper, I think that it is most appropriate to use this throughout.

Again, thank you. I hope my revisions will move the paper closer towards an acceptance.